# Gut microbiota diversity repeatedly diminishes over time following maintenance infliximab infusions in paediatric IBD patients

**Katrine Carlsen**[1,2☯]*, **Louise B. Thingholm**[3☯], **Astrid Dempfle**[4], **Mikkel Malham**[1,2], **Corinna Bang**[3], **Andre Franke**[3], **Vibeke Wewer**[1,2]

**1** Department of Paediatrics and Adolescence, Copenhagen University Hospital–Amager Hvidovre Hospital, Hvidovre, Denmark, **2** Copenhagen Center for Inflammatory Bowel Disease in Children, Adolescent and Adults, Copenhagen University Hospital, Amager and Hvidovre, Hvidovre, Denmark, **3** Institute of Clinical Molecular Biology, Christian-Albrechts-University of Kiel, Kiel, Germany, **4** Institut für Medizinische Informatik und Statistik, Universitätsklinikum Schleswig-Holstein, Christian-Albrechts-Universität zu Kiel, Kiel, Germany

☯ These authors contributed equally to this work.
* Katrine.carlsen@regionh.dk

**Data Availability Statement:** Raw microbiome data is available at EBI-ENA under PRJEB54570 with sample metadata. Full metadata on patients

## Abstract

### Background

The gut microbiome plays a crucial role in the pathogenesis and progression of inflammatory bowel disease (IBD). Understanding the dynamics of the gut microbiome in relation to treatment can provide valuable insights into disease management and therapy strategies. The aim of this study is to investigate if diversity and composition of the gut microbiome correlate with time since treatment and disease activity during maintenance infliximab (IFX) therapy among children with IBD.

### Methods

Data was collected from IBD patients aged 10–17 participating in an IFX-eHealth study. IFX infusions were administered in 4–12-week intervals based on weekly faecal calprotectin (FC) combined with symptom scores. Excess stool samples underwent microbiome profiling using 16S rRNA gene sequencing. Microbiome features, including alpha diversity and single taxa, were analysed for three key variables: 1) weeks-since-treatment, 2) FC, and 3) symptom score.

### Results

From 25 patients (median age 14.4 years) diagnosed with Crohn´s Disease (n = 16) or ulcerative colitis (n = 9), microbiota were analysed in 671 faecal samples collected across 15 treatment intervals.

A significant decrease over time in Shannon diversity, following the initial increase within four weeks of treatment, was found across patients. FC levels showed no association with alpha diversity (*p*>0.1), while symptom scores showed a negative association with Shannon and observed diversity in patients with UC. At the genus level, a lower abundance of the genera *Anaerostipes* and *Fusicatenibacter* (Firmicutes), and a greater abundance of the

cannot be shared publicly because of limitations enforced by the protocol approved by the ethics board. For metadata on the patients please inquire with Copenhagen IBD center, Amager Hvidovre University Hospital, Kettegaard alle 30, 2650 Hvidovre, tlf +45 38 62 38 62, https://www.hvidovrehospital.dk/cph-ibd

**Funding:** Louis-Hansens Foundation • URL: (https://louis-hansenfonden.dk) • NO - Include this sentence at the end of your statement: The funders had no role in study design, data collection and analysis, decision to publish, or preparation of the manuscript. • Initials of the authors who received the award: MM • Grant number: J.nr. 18-2B-2662 Deutsche Forschungsgemeinschaft (DFG) Research Unit 5042: miTarget - The Microbiome as a Therapeutic Target in Inflammatory Bowel Diseases. • Funding was given to a larger group of investigators as listed at https://www.mitarget.org/people/?filter-role=5. Leading investigator for this part is Prof. Andre Franke. Initials AF. • URL of funder website: https://www.dfg.de/en/ • NO - Include this sentence at the end of your statement: The funders had no role in study design, data collection and analysis, decision to publish, or preparation of the manuscript.

**Competing interests:** The authors have declared that no competing interests exist.

genus *Parasutterella* (Proteobacteria), were associated (*p*.adj<0.05) with the time elapsed since last infusion in UC specifically, while only *Parasutterella* was associated across the full cohort (p.adj = 1e-10).

## Conclusions

We found a recurring reduction over time in alpha diversity following the initial increase in diversity after an IFX infusion. Changes in an individual's microbiome may be an early sign of increasing disease activity that precedes clinical symptoms and increased FC.

## Introduction

Inflammatory bowel diseases (IBD), typically Crohn's Disease (CD) and ulcerative colitis (UC), are characterized by chronic inflammation of the gastrointestinal tract with alternating periods of remission and relapse. The incidence of paediatric-onset IBD (pIBD) is 8–15 per 100, 000 and is increasing globally for reasons not yet fully understood [1,2]. Approximately 20% of all IBD patients will develop their IBD during childhood or adolescence [3,4]. Symptoms includes stomach pain, diarrhoea, bloody stools, perianal symptoms, impaired growth, and delayed puberty, as well as symptoms of the liver (primary sclerosing cholangitis), eyes (uveitis, episcleritis), joints (arthralgia, arthritis) and skin (erythema nodosum, pyoderma gangrenosum). The aetiology and underlying factors contributing to disease development remain largely unknown. It is thought that the gut microbiome plays a key role in the development of IBD and impacts the disease course; however, despite extensive research during the last two decades the gut microbiome's precise role in IBD has not been determined [5–8]. Several studies have found that patients with IBD, in contrast to healthy controls, exhibit dysbiosis, i.e., a composition of the microbiome deviating from that observed in healthy controls, and especially lower gut diversity [9]. The dysbiosis appears to be associated with both IBD in general and with increased disease activity more specifically [6,10].

Some IBD treatments seek to enrich microbiome diversity and decreased mucosal inflammation achieved after induction of Infliximab (IFX) treatment among children with CD have associated with a shift of the faecal microbiome toward a healthier composition. However, results have varied as to how persistent this benefit is once remission has been achieved [11–16]. Recent studies among children with IBD have suggested that the composition of the gut microbiota before treatment can predict the IFX treatment response [17,18] which emphasize the need for further investigation of the role of the microbiome composition and response to treatment.

### Aim

The aim of this study was to use repeated follow-up sampling to investigate the diversity and composition of the gut microbiome over time and its correlation with time since IFX treatment and disease activity in moderate-to-severe pIBD patients undergoing maintenance therapy.

## Materials and methods

### Study subjects, sample collection and data overview

Longitudinal clinical data was collected from children from Zealand, Denmark, who were aged 10–17 years, and had been diagnosed with IBD and were undergoing treatment with IFX.

Recruitment and sample/data collection happened between September 1, 2013 and April 30, 2016 with a maximum follow-up of 2 years. All data was anonymised during analysis. Diagnoses of CD and UC were made according to the Porto criteria [19]. The indications for IFX treatment were an acute, severe disease with no response to intravenous corticosteroids, exclusive enteral nutrition, and/or moderate disease activity despite immunosuppressive treatment.

All patients had participated in an eHealth intervention study [20] between September 1, 2013 and April 30, 2016, with a maximum follow-up of two years, and all data and samples were originally collected as part of this study. The eHealth intervention, www.young.constant-care.com, has been described elsewhere [20,21], but briefly, it was an interactive tool for the patient and IBD team to assist in the timing of IFX infusions during maintenance treatment (IFX dose 5 mg/kg). Patients in the induction phase of IFX treatment (IFX 5 mg/kg week 0, 2, 6) were not included. Treatment intervals, with a minimum of four weeks and a maximum of twelve, were determined by the Total Inflammatory Burden Score (TIBS), which was based on the level of faecal calprotectin (FC), combined with a symptom score specifically designed for either UC or CD, the Paediatric Ulcerative Colitis Activity Index (PUCAI) [22] or the abbreviated Paediatric Crohn´s Disease Activity Index (abbrPCDAI) respectively [23]. The TIBS algorithm weighted both the degree of symptoms and the level of FC (score 0–10) and a TIBS >5 or a score between 2–4 two weeks in a row triggered an appointment at the hospital [24]. On a weekly basis, patients entered their symptoms and submitted a faecal sample by mail to the laboratory for the analysis of FC from four weeks after their last IFX infusion until the following treatment was initiated. FC levels were analysed at the time of sample collection and remaining faeces were stored at -20˚C for later microbiota analysis. FC analyses were performed using enzyme-linked immunosorbent assay (Calpro LtD., Lysaker, Norway). In summary, the eHealth study has built a longitudinal cohort based on weekly faecal samples and symptom scores collected between IFX infusions.

### Data generation and processing

Stool samples were subjected to library preparation and sequencing using a standardized protocol at the wet lab in Kiel, Germany [25].

PCR-based DNA amplification of the 16S rRNA gene region, V1-V2, was performed using the 27F-AGAGTTTGATCCTGGCTCAG/338R-TGCTGCCTCCCGTAGGAG primer combination. The PCR product was normalized using the SequalPrep Normalization Kit and sequencing was performed on the Illumina MiSeq platform, using the MiSeq Reagent Kit v3 according to the manufacturer's instructions. The following demultiplexing process was used to prevent mismatches in the index sequences: Data quality filtering and microbiome profiling was performed in the R software environment (v.3.5.1), using the DADA2 [26] (v.1.14) workflow for big datasets (https://benjjneb.github.io/dada2/bigdata.html), with the specific settings truncLen = c(260,200), maxN = 0, maxEE = c(2,2), truncQ = 5 and rm.phix = TRUE. Read-pairs that could not be merged due to insufficient overlap or mismatches were discarded and chimeras removed with method = "consensus". The resulting ASV abundance table was annotated using the Bayesian classifier provided with DADA2 and the Ribosomal Database Project v.6 release.

The workflow used can be found on GitHub: https://github.com/mruehlemann/german_mgwas_code/tree/master/1_preprocess.

Further details about the data and data generation can be found in the STORMS Microbiome Reporting Checklist (S1 Table).

## Data analyses and statistics

Data analysis was performed in the R software (v.4.0.4). The ASV table and taxonomic annotations table generated with DADA2 were joined to a phyloseq object and QC samples were removed. ASVs assigned to *Chloroplast or Mitochondria* were removed and the data filtered to a minimum sample read depth of 10,000 reads. The object was transformed both to relative abundances and rarefied to a minimum sample sum of 10,068 reads. Metadata were filtered to remove samples collected in an interval before or after the registered use of antibiotics and joined with phyloseq objects, resulting in objects with 748 samples. Data were further filtered to remove samples with missing information and any collected at a time after which no additional treatments were given, leaving 727 samples across 25 patients for further analysis (see Table 1). As the age span was limited and gender was balanced, these variables were not included in the statistical analyses (Table 1). For analysis of the association between

**Table 1. Patient demographics.**

|  | N = 25 |
|---|---|
| **Gender:** |  |
| Male, N (%) | 14 (56) |
| Female, N (%) | 11 (44) |
| **Disease:** |  |
| Crohn's disease, N (%) | 16 (64) |
| Ulcerative colitis, N (%) | 9 (36) |
| **Age at inclusion, years median (IQR)** | 14.4 (12.5;16.0) |
| **Age at diagnosis, years median (IQR)** | 12.0 (10.2;14.1) |
| **Duration from diagnosis to inclusion, years median (IQR)** | 1.5 (1.0;3.5) |
| **Disease extension:** |  |
| **Crohns's disease, N (%)** |  |
| Disease onset younger than 16 years, A1 | 16 (100) |
| Terminal ileum, L1 | 1 (6) |
| Colon, L2 | 4 (25) |
| Ileo colon, L3 | 6 (38) |
| Upper gastrointestinal disease, L4 |  |
| L1+L4 | 2 (13) |
| L2+L4 | 0 |
| L3+L4 | 3 (19) |
| P, Perianal disease modifier | 5 (31) |
| B1, Nonstricturing, nonpenetrating | 11 (69) |
| B2, Stricturing | 1 (6) |
| B3, Penetrating | 4 (25) |
| **Ulcerative colitis, N (%)** |  |
| E1, Proctitis | 0 |
| E2, Left-sided | 1 (11) |
| E3, Extensive | 8 (89) |
| **Number of patients with co-medication, N (%)** |  |
| +Thiopurines | 9 (36) |
| +Thiopurines and 5-ASA | 8 (32) |
| **Stool samples (N = 727):** |  |
| Median samples per patient (IQR) | 31 (10;43) |

IQR: Interquartile range; 5-ASA: 5-aminosalisylic acid.

microbiome and weeks since treatment, we further removed all treatment intervals where two or fewer samples had been collected. This resulted in the removal of 58 samples, leaving 671 samples for analysis. If a patient was also being treated with 5-aminosalisylic acid (5-ASA) or thiopurines (azathioprine or 6-mercaptopurine) during their treatment with IFX, this was recorded.

Alpha diversity measures were calculated using the phyloseq::estimaterichness function, where Shannon diversity was calculated using the unfiltered counts and observed ASVs estimated using the rarefied data. An overview of alpha diversity across the dataset is provided in supplementary data, S1 Fig. A mixed linear model was used to analyse the association of Shannon and observed richness with weeks since treatment using the lme4::lmer function. For analysis across all 25 subjects the model included a fixed effect interaction term for week and diagnosis, and a random term for subject, with interval nested in the subject. For the per-disease subtype analysis (analyses for CD and UC separately), we used a fixed effect term for week and a random effect term for subject. To assess whether 5-ASA and thiopurines affected alpha diversity, we used a cross-subject-like model, where the fixed effect was 5-ASA or thiopurines and weeks since treatment. Neither drug showed an association with Shannon or observed diversity ($p > 0.1$); consequently, neither drug was considered in the analysis of alpha diversity.

Permutational Multivariate Analysis of Variance Using Distance Matrices (permanova) was performed using the vegan [27]::adonis function, with Aitchison distances (method = "Euclidean" with CLR-transformed object (microbiome::transform)), 999 permutations within subject, including 671 samples and controlling for diagnosis (UC or CD) for the microbiome community at the genus level (phyloseq::tax_glom with taxrank = Genus and NArm = T). Analyses were performed for weeks since treatment, symptom score and square root-transformed FC. To consider the possible effects of 5-ASA and thiopurines, these drugs' association with overall microbiome composition was first examined in a permanova analysis. 5-ASA showed a significant association with microbiome composition ($R^2 = 0.04$, $p = 0.001$), while thiopurines showed no significant association ($R^2 = 0.027$, $p = 0.96$). Therefore, only 5-ASA were considered in the permanova analyses. Fore S2 Fig, showing the relationship between samples from five patients, Aitchison diversity was sued with genus level taxa and principal components ordination (phyloseq::ordinate with method = "RDA" and phyloseq:: plot_ordination).

For analysing associations between single taxa and weeks since treatment, FC or symptom score, negative binomial (nb) generalized linear mixed models (GLMM) were used for the 20 most abundant taxa at the genera and family levels. The threshold was set using genera with a minimum mean abundance of 1%. Setting this threshold was supported by visual inspection of single taxa behaviour over weeks since treatment for each patient (see example in S3 Fig).

To identify taxa that were associated with weeks since treatment or disease subtype, three models were designed for the analysis of each taxon. The first model assessed the association of the taxon abundance (dependent variable) with weeks since treatment and disease subtype (CD/UC), and included an interaction term between these two variables in the fixed effect term, in addition to an offset term for sequencing depth. This allowed us to identify where there was a significantly different association between weeks since treatment and taxon abundance among disease subtypes and, if there was no difference, draw a conclusion as to the association between taxon abundance and weeks since treatment across all CD and UC patients. The two further models each included only one of the subgroups of patients (CD or UC), with the disease subtype term removed. Together, these three models allowed us to identify 1) associations between taxa and weeks since treatment in CD and UC), 2) trends dependent on CD/ UC subtype, and 3) trends within CD or UC. 5-ASA, but not thiopurines, were included as a confounding variable in the analysis of single taxa based on the evaluations of alpha- and beta-

diversity, where 5-ASA were associated with beta-diversity. To adjust for testing of multiple taxa the stats::p.adjust function in R with method ="BH" was used and adjusted p-values given with "*p.adj*".

To identify taxa that were associated with FC or symptom score, a similar analysis was made; however, as we aimed to detect taxa associated with FC or disease score independent of development over time, the interval variables were removed from the random effect term. Furthermore, as FC measures had a long tail, with the bulk of the measurements close to zero, the variable was square root-transformed in order to stabilize the model.

Of the 732 samples, 110 did not include a recorded disease score and one did not include FC, meaning the power of the analysis of disease score is less than that for week since treatment (n = 671 for weeks since treatment and n = 616 for symptom score analyses).

### R and additional R packages used

vegan v2.5, ggplot2 v3.3, plyr v1.8.6, knitr v1.33, dplyr v1.0, RColorBrewer v1.1, gridExtra v2.3, grid v4, lattice v0.20, ggpubr v0.4, openxlsx v4.2, phyloseq v1.34, reshape2 v1.4, data.table v.1.14, tidyr v.1.2, NBZIMM v1.0, magrittr v2.0.

### Patient and public involvement

Patients or the public were not involved in the design, or conduct, or reporting, or dissemination plans of our research.

### Ethical considerations

This eHealth study was approved by the Danish Ethical Committee (jr.H-2-2013-061). All patients and their guardians gave oral and written consent before their inclusion in the study.

## Results

### Project overview

Twenty-nine patients were included in the eHealth study and the median follow-up was 553 days (IQR 217–696 days). Patient demographics are presented in **Table 1**. In total, 216 IFX treatments were given. The mean number of treatment pr. patient was 7.45 (SD 3.54). The mean interval between IFX treatments was 9.3 weeks (SD 1.9 weeks), and the indication for giving another infusion was based on TIBS score (FC/symptom score). The distribution of treatment intervals before and after 8 weeks was 26,7% before 8 weeks, 8,6% at week 8 and 64,7% after 8 weeks.

Data were filtered to remove samples with no associated microbiome data, no follow-up treatment, any samples collected after recent antibiotics, or any with missing data. This left 727 samples, comprising 16 CD patients with a combined 418 samples, and nine UC patients with a combined 309 samples. The 727 samples were collected across 15 treatment intervals, with two to 64 samples per patient. To analyse the association between microbiome and weeks since treatment, we further removed all intervals where two or fewer samples had been collected; this meant 58 samples were removed, leaving 671 samples available for analysis.

### Main outcome

The microbiome was analysed for three key variables: 1) weeks since treatment; 2) FC, and 3) symptom score. FC and symptom score were taken to be indicative of disease activity (where higher levels of FC and higher symptom scores represented increased disease activity). Both FC and symptom score were associated with weeks since treatment (GLMM nb model, FC

p = 6.5e-14 beta = 0.013; symptom score p = 7.9e-5 beta = 0.02). In total, 126 treatments are represented in the data set and across 20 intervals and the maximum number of weeks without treatment was twelve. FC levels of samples were distributed as follows: 69% were less than 250 mg/kg, 19% were 250–749 mg/kg, and 11% were higher than 749 mg/kg.

## Changes in microbiome diversity in the lead-up to treatment

Fig 1 illustrate the bacterial community from one patient during twelve treatment periods.

The two measures of within-sample microbiome diversity (alpha diversity)–Shannon diversity and the number of observed bacteria (also called 'richness')–were assessed for their association with the weeks leading up to the next maintenance treatment. Across all samples, Shannon diversity showed a significant decrease as the time elapsed since the last treatment (lmer model, beta = -0.018, $p$ = 0.036), and the per disease subtype analysis found a similar, but non-significant, negative trend in CD patients (lmer beta = -0.017, $p$ = 0.057), but no such association in UC (lmer beta = -0.009, $p$ = 0.38). Observed taxa across subjects showed no association with the weeks since last treatment, while a trending but non-significant association appeared within the subset of UC patients (lmer beta = -2.16, $p$ = 0.07, **Fig 2**).

For FC and symptom scores, a visual inspection indicated limited linear relationships between alpha diversity and these two individual measures (see **S5 Fig** for a visualization of Shannon diversity), and this observation was supported by statistical analyses using the linear mixed model described above. FC levels showed no association with either alpha diversity (glmm $p$>0.1 for all models, including all patients and when testing disease subgroups (UC and CD) separately), while symptom score showed a negative association with Shannon and observed diversity in UC patients (glmm Shannon, $p$ = 0.012 and observed, $p$ = 0.0007).

Patients were given a new IFX infusion either as a result of a relapse based on their TIBS score (increased symptom score and/or FC) or after a maximum of 12 weeks without treatment. As such, intervals were sub-grouped into those ended by relapse and those ended after reaching the 12-week limit. We evaluated whether there was a difference in the development of alpha diversity across weeks since treatment between these two subgroups. Patients who needed treatment before week 12 due to increased disease activity had lower alpha diversity, which also appeared to decrease further as the time elapsed since the last treatment, compared to patients in remission during the 12-week IFX interval (see **Fig 2C and 2D**). However, statistical analysis did not detect a significant difference in the relationship between alpha diversity and weeks since treatment between the two groups (mixed model analysis, adjusting for patient dependence with random term for patient, $p$>0.1).

Analysis of the overall community composition (beta-diversity) and weeks-to-treatment identified a significant association (permanova analysis using vegan::adonis function with Aitchison distances, including 671 samples and controlling for diagnosis (CD or UC) and 5-ASA; $R^2$ = 0.004, $p$ = 0.047). Correspondingly, both symptom score and FC were associated with overall community composition (symptom score $R^2$ = 0.011, $p$ = 0.018; FC $R^2$ = 0.018, $p$ = 0.001). As such, analysis of the overall community composition supported an association between the microbiome composition and the outcome measures, with the alpha diversity analyses showing that the gut microbial diversity becomes less diverse as the time elapsed since the last treatment increases. Visual inspection of the overall community composition (using principal components ordination) showed a general clustering of samples from the same subject, as expected, based on prior analysis of human gut microbiomes across individuals (see **S2 Fig** which show clustering by subject for a selected subset of five patients).

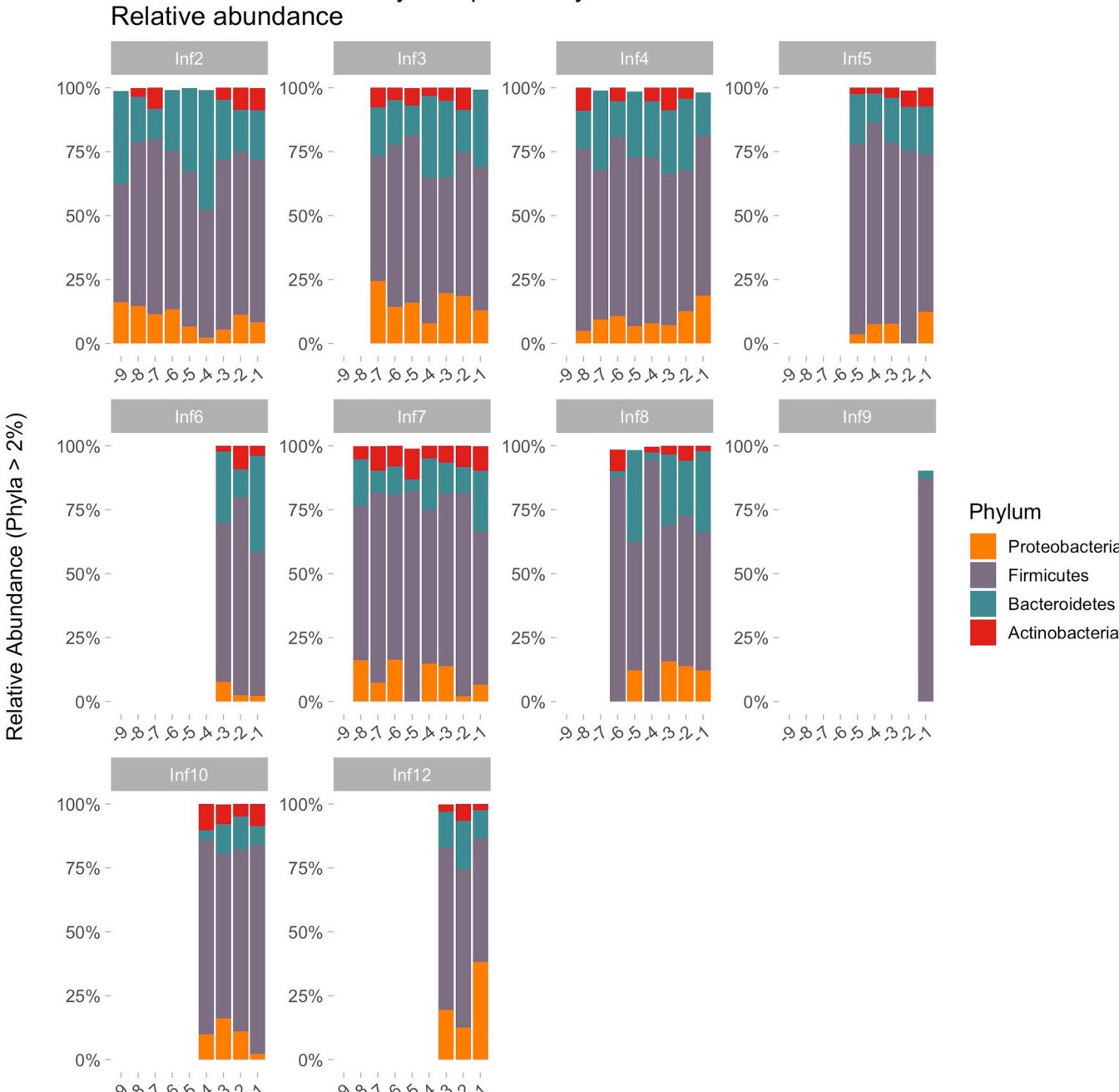

**Fig 1. Microbiome profile of the most abundant phyla in one patient (Patient 1), across 12 infliximab treatments (-1 is the week before treatment, -2 the two weeks before treatment, etc).** The figure includes only phyla with min. 2% relative abundance to properly show distinguishable colours and provide a readable legend.

### Single taxa analysis

To identify single taxa associated with either weeks since treatment, FC, or disease score, negative binomial generalized linear mixed models were used to analyse the top 20 most abundant taxa at the genera and family levels. For each single taxa three models were applied, one

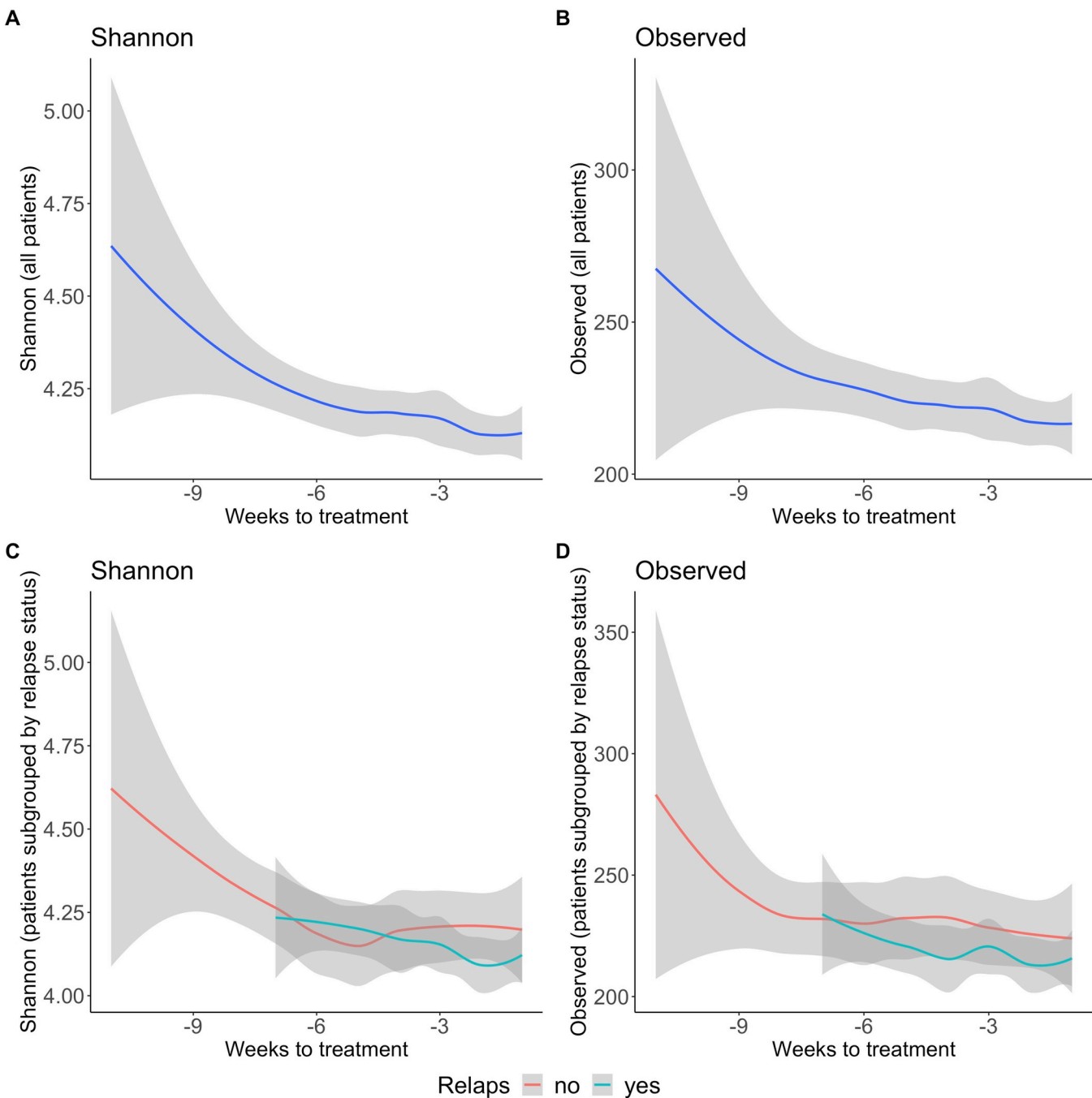

**Fig 2. Changes in alpha diversity in the weeks before anti-TNF-a treatment.** The two alpha diversity measures, Shannon (**A, C**) and observed (**B, D**), were analysed for their association with weeks-to-treatment (mixed linear model, based on 671 samples and 25 patients, see Methods). Analyses were performed across all patients shown here (**A-B**) and within the two disease subtypes, as shown in **S4 Fig** (9 UC and 16 CD). The analysis found a significant association across all subjects for Shannon diversity with decreasing diversity in the lead up to the next treatment (lmer model, beta = -0.018, $p = 0.036$) and no association for observed diversity ($p>0.1$). The bottom plots (**C-D**) show the trend for alpha diversity across weeks to treatment, when patients were sub-grouped according to those who suffered a relapse and those who were treated with anti-TNF-a because twelve weeks had elapsed since their last treatment. The relapse group showed a lower diversity that continued to fall prior to the next treatment, but the analysis detected no significant difference in the relationship between weeks-to-treatment and alpha diversity across the two patient subgroups (lmer, $p>0.1$).

analysing all patients while considering a possible effect of CD/UC subtype, and two models within CD and UC subgroups. This allowed us to identify associations of single taxa with phenotype (weeks since treatment, FC, or disease score), while considering the possible differences between CD and UC. Summaries of these statistics can be found in **S2 and S3 Tables**. A taxonomy of the genera we analysed are presented in **Table 2**.

## Analysis of time (weeks) since treatment for single taxa

Prior studies have identified differences in the gut microbiome between CD and UC, and it is therefore important to consider a possible role for disease subtype in the current cohort. At the genus level, there was a significantly different association between taxa and weeks since treatment between the two disease subtypes (GLMM, *p.adj* = 0.001) for *Parasutterella*, while there was a nominal difference for *Sutterella* (GLMM, *p* = 0.049) and no significant difference for *Faecalibacterium* (GLMM, *p* = 0.062). For the taxa of interest it is necessary to pay attention to the within-disease subtype results. At the family level, *Pasteurellaceae* showed a different

**Table 2. Study taxonomy.** Families and genera included in the analyses are marked in bold.

| Domain | Phyla | Class | Order | Family | Genus |
|---|---|---|---|---|---|
| Bacteria | Actinobacteria | Coriobacteriia | Coriobacteriales | **Coriobacteriaceae** | **Collinsella** |
| | Bacillota | Bacilli | Lactobacillales | **Lactobacillaceae** | |
| | Bacteroidetes | Bacteroidia | Bacteroidales | **Bacteroidaceae** | **Bacteroides** |
| | | | | **Prevotellaceae** | **Prevotella** |
| | | | | **Rikenellaceae** | **Alistipes** |
| | | | | Tannerellaceae | **Parabacteroides** |
| | Firmicutes | Bacilli | Lactobacillales | **Streptococcaceae** | |
| | | Clostridia | Clostridiales | **Ruminococcaceae** | |
| | | | Eubacteriales | Clostridiaceae | **Clostridium** |
| | | | | **Clostridiaceae_1** | |
| | | | | **Clostridiales_Incertae_Sedis_XIII** | |
| | | | | **Lachnospiraceae** | **Anaerostipes** |
| | | | | | **Blautia** |
| | | | | | **Fusicatenibacter** |
| | | | | Oscillospiraceae | **Faecalibacterium** |
| | | | | | **Oscillibacter** |
| | | | | | **Ruminococcus** |
| | | | | | **Subdoligranulum** |
| | | | | **Peptostreptococcaceae** | **Romboutsia** |
| | | Erysipelotrichia | Erysipelotrichales | **Erysipelotrichaceae** | |
| | | Negativicutes | Acidaminococcales | **Acidaminococcaceae** | |
| | | | Veillonellales | **Veillonellaceae** | **Dialister** |
| | Fusobacteria | Fusobacteriia | Fusobacteriales | **Fusobacteriaceae** | |
| | Proteobacteria | Betaproteobacteria | Burkholderiales | **Sutterellaceae** | **Parasutterella** |
| | | | Burkholderiales | | **Sutterella** |
| | | Gammaproteobacteria | Enterobacterales | **Enterobacteriaceae** | **Escherichia** |
| | | | | Morganellaceae | **Proteus** |
| | | | Pasteurellales | **Pasteurellaceae** | |
| | | | | **Porphyromonadaceae** | |
| | Verrucomicrobia | Verrucomicrobiae | Verrucomicrobiales | Akkermansiaceae | **Akkermansia** |
| | | | | **Verrucomicrobiaceae** | |

association with weeks since treatment between the CD and UC disease groups (GLMM, *p. adj*<0.1), while no other families showed an association with p.adj<0.1 in the GLMM analyses.

*Anaerostipes*, *Fusicatenibacter*, and *Parasutterella* showed a significant association with weeks since treatment within the UC subtype (GLMM, *p.adj* = 0.037, 0.037 and 0.037, respectively, **Fig 3** and **S2 Table**). *Anaerostipes* and *Fusicatenibacter* had a negative association (decreasing abundance prior to the next treatment), while *Parasutterella* had a positive association (increasing abundance prior to the next treatment). All three genera also showed a nominal association in CD and across all patients (GLMM, *p*<0.1). *Parasutterella* had a significant interaction term as given above, meaning that the association between abundance and weeks since treatment was found to be different between the CD/UC subtypes; however, CD showed a trending non-significant that was positive like that observed for UC (GLMM, *p* = 0.057). No further genera showed significant associations with weeks since treatment (glmm, *p.adj*<0.05). Above the significance threshold in the GLM analyses were *Akkermansia* (positive beta value in UC), *Blautia* and *Clostridium* sensu stricto (with a negative beta in CD and all patients in total), *Prevotella* (with a positive beta in CD and all patients), *Subdoligranulum* (with a positive beta in CD) and, lastly, *Sutterella* (with a positive beta in CD and all patients, but also a nominal interaction) (**Fig 4** and **S2 Table**). Family-level results are presented in **S3 Table** and **S6 Fig**. As for weeks since treatment, we analysed the association of single taxa with FC and symptom score and the genus-level results are shown in **Fig 4** and **S2 Table**, and family clades results in **S6 Fig** and **S3 Table**.

## Discussion

In this longitudinal study of 671 faecal samples collected between IFX infusions in 25 paediatric IBD patients, we found a decrease in alpha diversity as the time elapsed since the last treatment. We found no association between levels of FC and alpha diversity; however, symptom scores showed a negative association with alpha diversity in UC patients. At the taxonomic

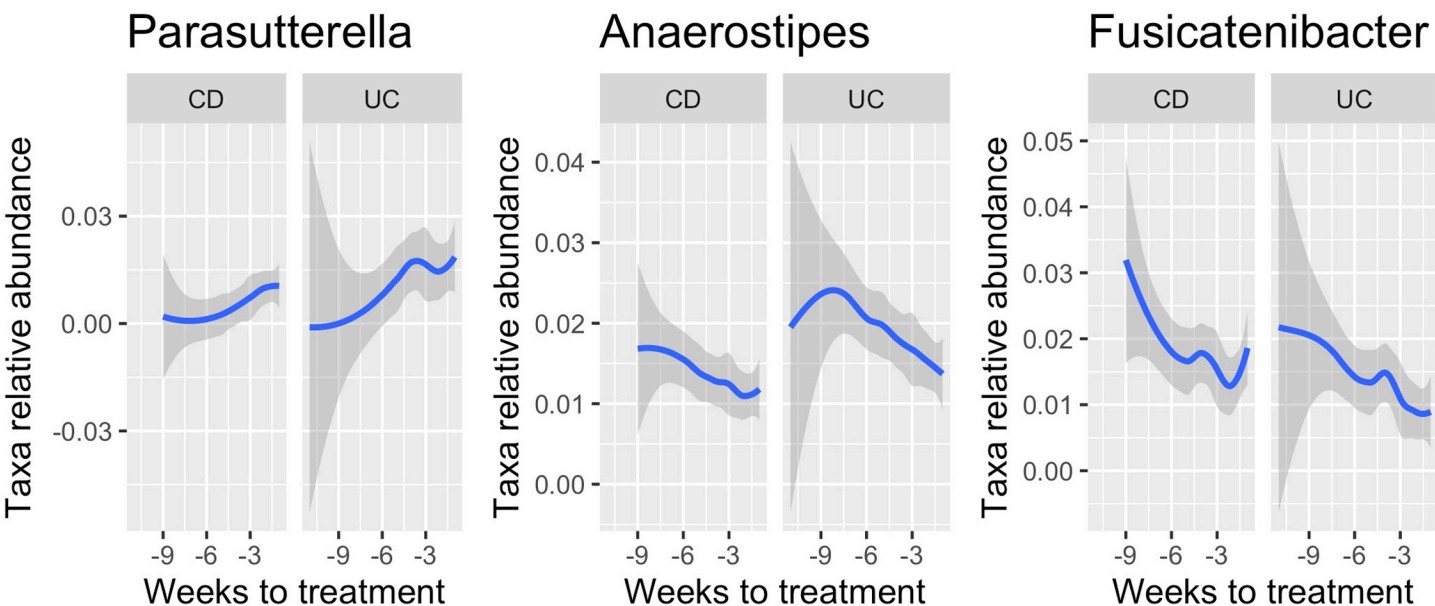

**Fig 3. Association across weeks to treatment for three genera.** *Parasutterella*, *Anaerostipes* and *Fusicatenibacter* were all found to be associated with weeks to treatment. The panels show a smooth trend line across weeks to treatment for both CD and UC subjects.

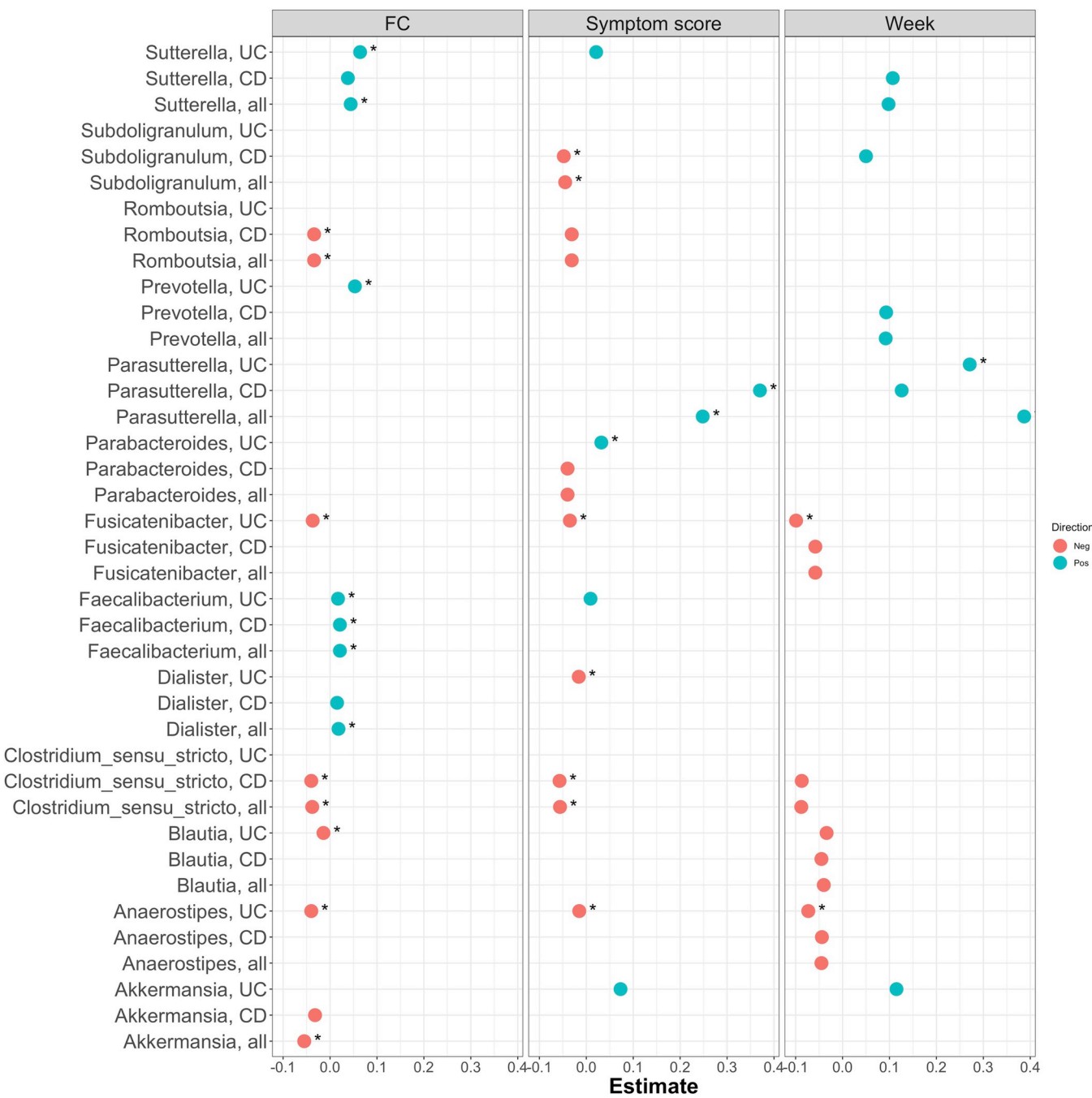

**Fig 4. Genera associated with FC, symptom score or week to treatment.** Each genera was analysed in a model that included all patients, as well as separate models for each patient subgroup. The figure shows all bacteria associated with either weeks since treatment, symptom score or FC (*p.adj*<0.1). For each bacterium reaching the threshold of significance, all three subgroups were retained (CD, UC, and the two combined), and a dot indicates where the nominal *p*-value is <0.1. Associations remaining significant after multiple-testing adjustment, at *p.adj*<0.1, are marked with an asterisk. The x-axis shows the estimated coefficient, with positive values indicating an increase in taxa abundance toward next treatment (weeks), or a positive association with symptom score or FC. For family-level results see **S5 Fig**.

level, we found an increased abundance of the Proteobacteria *Parasuterella*, and a lower abundance of the Firmicutes *Fusicatenibacter* and *Anaerostipes*, associated with time since last treatment.

## Alpha diversity

The observed diversity dynamics of this study likely reflect the concentration of IFX that will initially increase and then decrease over time after infusion, and consequently, the anti-inflammatory effect of the treatment, and thus the associated change in the microbiome, will depend on time since treatment.

Dysbiosis of the gut microbiome is related to IBD, and a wide range of studies have pointed to associations between dysbiosis and increased disease activity in both CD and UC. However, study results about how the microbiome changes, and how long changes persist after initiating treatment, vary widely. In a study of 110 pIBD patients, Olbjørn et al. [12] used a targeted approach with predefined bacteria to characterize gut dysbiosis and reported higher rates of dysbiosis in their patients than in healthy controls. Of these patients, those given biological therapy had an even lower abundance of Firmicutes, and CD patients with complicated disease behaviour had a higher abundance of Proteobacteria. Regardless of the type of treatment or whether patients achieved mucosal healing, Olbjørn et al. found no change in the microbiome composition after 18 months of follow-up. In contrast, Kowalska-Duplaga et al. [13] found no significant differences in alpha diversity when comparing healthy children and paediatric CD patients after induction of IFX therapy.

In another study of paediatric CD patients undergoing IFX treatment, Wang et al. [11] found that the diversity of the gut microbiota improved after treatment had been initiated and that the bacterial community closely resembled that of healthy controls. In this study, only four patients out of 11 had a sustained therapeutic response to IFX, and they observed that short-chain-fatty-acids (SCFA)-producing taxa were more abundant than in patients with a non-sustained therapeutic response. An abundance of SCFA-producing bacteria has also been confirmed in a more recent study by the same group [14].

Results from a study including children with CD and juvenile arthritis investigating the microbiome before and after induction with IFX, concluded that a shift in the composition of the faecal microbiome were related to a less inflamed gut mucosa rather than general effect of IFX [16]. These results suggest that the burden of inflammation drive the microbiome composition regardless category of treatment.

According to the design of our eHealth study, patients were expected to experience symptoms and/or increased levels of FC, or to reach the maximum treatment-free period of 12 weeks, prior to scheduling their next treatment. Even considering those patients who had a sustained response and remission after IFX (during a 12-week interval), we observed overall lower gut diversity prior to treatment (show in **Fig 2**). We interpret the observation that alpha-diversity first increases and then, as time since treatment increases, as a reflection of a fast response by the microbiome, either directly to the IFX treatment or indirectly to an improvement of the gut, that then diminishes. Our data illustrate that the microbiome was dynamic and alterable, and our data do not, therefore, support that increased diversity persists despite continued IFX treatment.

## FC levels and alpha diversity measure

We found no association between levels of FC and alpha diversity in CD and UC, which was surprising as higher FC indicates increased disease activity. Olbjørn et al. [12] found a higher abundance of Proteobacteria (indicating dysbiosis) in patients with FC greater than 1,000 mg/kg and Malham et al. [10] and Schirmer et al. [28] also found a reduced diversity in paediatric UC patients, as well as increased FC. One possible reason for our finding may be that the majority of the FC levels recorded in our study were below 750 mg/kg (89% of FC values were less than 750 mg/kg). Rajca et al. [15] showed that in adults with CD a low abundance of

*Faecalibacterium* p*rausnitzii* (Firmicute) could predict relapse one year after withdrawal of IFX, but they found no association between dysbiosis and FC. This prompts the question of whether dysbiosis is a cause of increased inflammation or, rather, that dysbiosis accelerates existing inflammation and could be the first sign of increased disease activity.

## Single taxa analysis

Studies comparing the gut microbiome of IBD patients and healthy persons have shown that IBD patients presents lower abundance of Firmicutes and increased abundances of Proteobacteria [6,29]. We also found a lower abundance of *Anaerostipes* (Firmicutes) and *Fusicatenibacter* (Firmicutes), and a higher abundance of the Proteobacteria *Parasuterella*, was significantly correlated with the time since the last IFX infusion in UC (p.adj<0.05) while only the association for *Parasuterella* was significant in the full cohort (p.adj = 1e-10). *Anaerostipes* are bacteria known to produce short chain fatty acids (SCFA) by fermentation in the luminal tract [6]. SCFAs are important for maintaining intestinal homeostasis [30]. The role and impact of SCFAs is complex and has been described as an intermediary between the microbiota and the immune system however findings on faecal SCFA in children with IBD have not been consistent [7]. SCFAs, primarily butyrate, are used as an energy source in intestinal epithelial cells, play a role in the inhibition of pro-inflammatory mediators, and are expected to increase anti-inflammatory mediators, as well as help to maintain the epithelial barrier function [9,31]. Accordingly, loss of SCFA-producing bacteria exacerbates intestinal inflammation in IBD patients. It is therefore prudent to continue investigating the possible benefits of faecal microbiota transplantation and the use of pre- and probiotics as part of IBD treatment [32].

## Strengths and limitations

The main strength of this study is its longitudinal follow-up during IFX treatment, with samples collected on a weekly basis. The considerable quantities of data we collected further enrich our knowledge of the microbiome in children with IBD. A second strength is that, currently, patients are usually treated with IFX every eight weeks, and intervals of up to 12 weeks are no longer advised [33–35]. As such, the data from this study are unique in that they are based on treatment intervals that are longer (up to a maximum of 12 weeks) than what are typically used in clinical practice today.

However, this study is limited by its small number of patients. Furthermore, some of these patients were not receiving IFX as a monotherapy, and their other treatments may have affected the microbiome. We were not in possession of information about the participants diet and lifestyle factors which may represent another limiting in the interpretation of the data.

The faecal samples were mailed by patients to the laboratory at RT, so the samples were exposed to fluctuating temperatures during shipment before entering storage at -20°C in the laboratory's freezer. This limitation was deemed acceptable however, considering both that samples were shipped over a short distance (from residence to local hospital), with robustness achieved from the high number of samples that were all subjected to a consistent storage and subsequent laboratory processing, and supported by prior benchmarking studies that indeed find shipment at RT less optimal however with high ICC for most diversity measures and single taxa evaluated [36].

Finally, we chose to analyse the 20 most abundant bacteria at the genera level and the 20 most abundant bacteria at the family level. We restricted the analysis to these bacterial taxa to reduce the burden of multiple testing and respect the power limitations inherent in the complex mixed models necessary to consider both subject and interval structure. As a result, we were only able to detect associations for these particular bacterial clades and were blind to

other possible associations with unselected bacteria. The clades we analysed are listed in **Table 2**, where their taxonomic relationships are shown.

## Conclusion

In our study of longitudinal data based on 671 faecal samples from 25 paediatric IBD patients we found that gut diversity diminished the longer it had been since the last IFX infusion. Our results suggest that increased disease activity impacts the microbiome by lowering the abundance of SCFA-producing bacteria. While we found no association between low diversity and FC, this may be explained by the somewhat low FC levels in the faecal samples we analysed, or that a decrease in diversity occurs before levels of FC increase. In conclusion, changes in an individual's microbiome may be an early sign of increasing disease activity that precedes other symptoms and increased FC.

## Supporting information

**S1 Table. STORMS microbiome reporting checklist.**
(XLSX)

**S2 Table. Summary statistics for analysis of single genera clades.**
(XLSX)

**S3 Table. Summary statistics for analysis of single family clades.**
(XLSX)

**S1 Fig. Alpha diversity measures across all 727 samples.**
(JPG)

**S2 Fig. Community composition of samples from five individuals show clustering of samples according to individual.**
(JPG)

**S3 Fig. Abundance of the top 20 most abundant genera in one patient (Patient 1), across the weeks running up to treatment.**
(JPG)

**S4 Fig. Development of alpha diversity across weeks towards a new TNF treatment for each IBD disease subtype.**
(JPG)

**S5 Fig.** Development of Shannon diversity across symptom scores (A) and FC values (B).
(JPG)

**S6 Fig. Family clades associating with week to treatment, symptom score or FC.**
(JPEG)

## Acknowledgments

We would like to thank Ilona Urbach, Ines Wulf, and Tonio Hauptmann of the IKMB microbiome laboratory, and the staff of the IKMB sequencing facilities, for their excellent technical support. We would further like to thank Malte Rühlemann for establishing the DADA2-based pipeline for 16S rRNA gene processing that was used in this paper. Thanks are owed to Magnus Lydolph of the Statens Serum Institute, Denmark, for supporting the logistics of handling the samples.

## Author Contributions

**Conceptualization:** Katrine Carlsen, Mikkel Malham, Vibeke Wewer.

**Formal analysis:** Louise B. Thingholm.

**Funding acquisition:** Mikkel Malham, Andre Franke.

**Methodology:** Corinna Bang.

**Project administration:** Katrine Carlsen, Corinna Bang.

**Supervision:** Astrid Dempfle, Andre Franke, Vibeke Wewer.

**Visualization:** Louise B. Thingholm.

**Writing – original draft:** Katrine Carlsen, Louise B. Thingholm.

**Writing – review & editing:** Mikkel Malham, Corinna Bang, Andre Franke, Vibeke Wewer.

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
