## [Decision Letter · Decision Letter 0]

15 May 2024

PONE-D-23-19986Gut microbiota diversity repeatedly diminishes over time following maintenance infliximab infusions in paediatric IBD patientsPLOS ONE

Dear Dr. Thingholm,

Thank you for submitting your manuscript to PLOS ONE. After careful consideration, we feel that it has merit but does not fully meet PLOS ONE’s publication criteria as it currently stands. Therefore, we invite you to submit a revised version of the manuscript that addresses the points raised during the review process. **Please address the reviewer comments by providing the requested clarifications as well as additional beta diversity analysis requested by Reviewer #2, and by adjusting the interpretations and Discussion to address study limitations.**

We look forward to receiving your revised manuscript.

Kind regards,

Jonathan Jacobs

Academic Editor

PLOS ONE

Journal Requirements:

1. When submitting your revision, we need you to address these additional requirements. Please ensure that your manuscript meets PLOS ONE's style requirements, including those for file naming. The PLOS ONE style templates can be found at https://journals.plos.org/plosone/s/file?id=wjVg/PLOSOne_formatting_sample_main_body.pdf and https://journals.plos.org/plosone/s/file?id=ba62/PLOSOne_formatting_sample_title_authors_affiliations.pdf 2. Did you know that depositing data in a repository is associated with up to a 25% citation advantage (https://doi.org/10.1371/journal.pone.0230416)? If you’ve not already done so, consider depositing your raw data in a repository to ensure your work is read, appreciated and cited by the largest possible audience. You’ll also earn an Accessible Data icon on your published paper if you deposit your data in any participating repository (https://plos.org/open-science/open-data/#accessible-data). 3. Thank you for stating the following in the Acknowledgments Section of your manuscript: We would like to thank Ilona Urbach, Ines Wulf, and Tonio Hauptmann of the IKMB microbiome laboratory, and the staff of the IKMB sequencing facilities, for their excellent technical support. We would further like to thank Malte Rühlemann for establishing the DADA2-based pipeline for 16S rRNA gene processing that was used in this paper. Thanks are owed to Magnus Lydolph of the Statens Serum Institute, Denmark, for supporting the logistics of handling the samples. This study was supported by the Louis-Hansens Foundation and the Deutsche Forschungsgemeinschaft (DFG) Research Unit 5042: miTarget - The Microbiome as a Therapeutic Target in Inflammatory Bowel Diseases.  We note that you have provided funding information that is not currently declared in your Funding Statement. However, funding information should not appear in the Acknowledgments section or other areas of your manuscript. We will only publish funding information present in the Funding Statement section of the online submission form. Please remove any funding-related text from the manuscript and let us know how you would like to update your Funding Statement. Currently, your Funding Statement reads as follows: Louis-Hansens Foundation •
URL: (https://louis-hansenfonden.dk) •
NO - Include this sentence at the end of your statement: The funders had no role in study design, data collection and analysis, decision to publish, or preparation of the manuscript.•
Initials of the authors who received the award: MM•
Grant number: J.nr. 18-2B-2662Deutsche Forschungsgemeinschaft (DFG) Research Unit 5042: miTarget - The Microbiome as a Therapeutic Target in Inflammatory Bowel Diseases.•
Funding was given to a larger group of investigators as listed at https://www.mitarget.org/people/?filter-role=5. Leading investigator for this part is Prof. Andre Franke. Initials AF.•
URL of funder website: https://www.dfg.de/en/•
NO - Include this sentence at the end of your statement: The funders had no role in study design, data collection and analysis, decision to publish, or preparation of the manuscript. Please include your amended statements within your cover letter; we will change the online submission form on your behalf. 4. In the online submission form, you indicated that Raw microbiome data are available at EBI-ENA under PRJEB54570 with relevant metadata. For additional metadata on the patients please inquire with corresponding authors. All PLOS journals now require all data underlying the findings described in their manuscript to be freely available to other researchers, either a. In a public repository, b. Within the manuscript itself, or c. Uploaded as supplementary information.This policy applies to all data except where public deposition would breach compliance with the protocol approved by your research ethics board. If your data cannot be made publicly available for ethical or legal reasons (e.g., public availability would compromise patient privacy), please explain your reasons on resubmission and your exemption request will be escalated for approval.  5. When completing the data availability statement of the submission form, you indicated that you will make your data available on acceptance. We strongly recommend all authors decide on a data sharing plan before acceptance, as the process can be lengthy and hold up publication timelines. Please note that, though access restrictions are acceptable now, your entire data will need to be made freely accessible if your manuscript is accepted for publication. This policy applies to all data except where public deposition would breach compliance with the protocol approved by your research ethics board. If you are unable to adhere to our open data policy, please kindly revise your statement to explain your reasoning and we will seek the editor's input on an exemption. Please be assured that, once you have provided your new statement, the assessment of your exemption will not hold up the peer review process. 6. Your ethics statement should only appear in the Methods section of your manuscript. If your ethics statement is written in any section besides the Methods, please move it to the Methods section and delete it from any other section. Please ensure that your ethics statement is included in your manuscript, as the ethics statement entered into the online submission form will not be published alongside your manuscript.  

Reviewers' comments:

Reviewer's Responses to Questions

**Comments to the Author**

1. Is the manuscript technically sound, and do the data support the conclusions?

Reviewer #1: Partly

Reviewer #2: Yes

2. Has the statistical analysis been performed appropriately and rigorously? 

Reviewer #1: N/A

Reviewer #2: Yes

3. Have the authors made all data underlying the findings in their manuscript fully available?

Reviewer #1: Yes

Reviewer #2: Yes

4. Is the manuscript presented in an intelligible fashion and written in standard English?

Reviewer #1: Yes

Reviewer #2: Yes

5. Review Comments to the Author

**Reviewer #1:** The study itself is interesting because of its longitudinal design and a decent amount of samples for the analysis. However, it is clearly visible that the study was originally done many years ago, and therefore, many caveats in regard to the microbiome analysis occurred. Mostly importantly, it’s the time from sampling to deep freezing of stool samples. Sending samples by mail is ok for calprotectin analysis, but it’s definitely worthless for microbiome testing if not sent on dry ice or at least in the cooling container, which was not specified. Squeezing another paper from an otherwise overwhelming sample collection and a great study might, therefore, not be impossible.

Also, there are a few minor issues in terms of the terminology of microbiome analysis and interpreting the p-values over 0.05 as significant, as seen in the table. Also, the literature review seems insufficient, as a few recent papers from respected journals are missing in the Introduction and Discussion.

In summary, I would strongly argue for clarifying my above-mentioned concerns and adjusting the paper in terms of the issues mentioned below. As the row numbering was surprisingly not present, I only referred to the page of the PDF I received.

The reviewer’s comments are mentioned in detail in the attached PDF.

**Reviewer #2: **The manuscript presents a longitudinal study investigating the changes in the alpha diversity of gut

microbiome in pediatric inflammatory bowel disease (IBD) patients in relation to infliximab (IFX)

treatment, fecal calprotectin (FC) levels, and symptom scores. The longitudinal approach of the study is

a key strength, as it provides insights into the dynamic nature of the gut microbiome in the context of

IBD management. Further, the authors employed appropriate statistical methods, including mixed linear

models, PERMANOVA, and negative binomial generalized linear mixed models, to analyze the

microbiome data and account for potential confounding factors. This rigorous analytical approach

strengthens the reliability of the findings. This reviewer suggest addressing a few concerns to make the manuscript strong.

Major concern:

The authors have mostly focused on alpha diversity assessment in these samples, which

provides a limited understanding of the microbiome changes. It would be beneficial to include

more information on beta diversity to capture the overall community composition changes in

relation to IFX treatment, FC levels, and symptom scores.

The authors should provide a justification for the observed pattern of alpha diversity increasing

in the first 4 weeks after IFX treatment but decreasing thereafter. A more in-depth discussion of

the potential mechanisms and implications would strengthen the manuscript.

The authors should clearly state the IFX dosing protocol used in the study, as this information is

crucial for interpreting the results. Additionally, the authors should discuss the potential impact

of the variable treatment intervals on the microbiome analysis. It would be helpful to know if

the microbiome changes observed were consistent across the different treatment intervals or if

there were any differences based on the duration between IFX infusions.

The authors mentions that treatment intervals with IFX were determined by the Total

Inflammatory Burden Score (TIBS), which was based on fecal calprotectin (FC) levels and

symptom scores, and the intervals ranged from a minimum of 4 weeks to a maximum of 12

weeks. However, it's not clear how many patients out of 25 received the IFX treatment at what

week interval; the 4 – 12 weeks' timeframe is broad. It will be helpful to include this

information and if some patients received multiple IFX infusions in the study, then it should be

described appropriately.

The authors should explain how the Total Inflammatory Burden Score (TIBS) (FC/symptom

score) was calculated. It looks like for the symptom scores, two different indices were used for

UC and CD patients, and additional information may help in understanding the scoring system.

The authors have accounted for the confounding effects of treatment with 5-aminosalicylic acid

(5-ASA) and thiopurines. However, the study did not account for other factors that could

influence the gut microbiome, such as diet and lifestyle factors. The effect of such factors on

the microbiome should be acknowledged in the discussion.

Minor concerns:

The abstract should provide a more comprehensive background and context for the study, rather than

simply stating the aim. A brief introduction to the relevance of the gut microbiome in inflammatory

bowel disease (IBD) and the importance of understanding its dynamics in relation to treatment would

help readers better appreciate the significance of the research question being addressed.

Please provide the primer sequences used to amplify V1V2 region of 16S rRNA gene in this study.

Please explain which 16S database was used for taxonomic assignments.

PCoA plot with just five representative samples is Okay (figure S3), but specifically mention this in the

manuscript.

The authors should consistently use appropriate "genus-level" or "genera level" taxonomic

identifiers throughout the manuscript.

In the discussion section "FC levels and alpha diversity measure" the name of the species and

the phylum is misspelled, it should be Faecalibacterium prausnitzii (Firmicute).

Figures:

Figure 1 is not referred to in the result section. The authors should either refer to this figure or

consider removing it if it is not essential.

For Figure 1, the authors should explain why phyla with over 2% relative abundance were

considered.

Figures 2A and 2B are not referred to in the result section. The authors should either refer to

these figures or consider removing them if they are not essential.

Figure 4 is not referred to in the result section. Also, the Y-axis title should be 'Taxa relative

abundance' (there is a typo).

Please provide the legends for figure S3.

For Figures 3 and S5, please add the legends within the figures.

Supplementary tables 1, 2, and 3 could not be found. The authors should provide all

supplementary materials and properly referenced them in the main text.

6. PLOS authors have the option to publish the peer review history of their article (what does this mean?). If published, this will include your full peer review and any attached files.

Reviewer #1: **Yes: **Jakub Hurych

Reviewer #2: No

---

## [Author Response · Author response to Decision Letter 0]

1 Jul 2024

Out response has been uploaded as it exceeds the character limit.

---

## [Decision Letter · Decision Letter 1]

1 Aug 2024

PONE-D-23-19986R1Gut microbiota diversity repeatedly diminishes over time following maintenance infliximab infusions in paediatric IBD patientsPLOS ONE

Dear Dr. Thingholm,

Thank you for submitting your manuscript to PLOS ONE. After careful consideration, we feel that it has merit but does not fully meet PLOS ONE’s publication criteria as it currently stands. Therefore, we invite you to submit a revised version of the manuscript that addresses the points raised during the review process.

The revised manuscript addressed all the reviewers’ initial comments. There was only a single comment by Reviewer #1 about the reporting of significant taxa associated with time elapsed since last infusion. I believe that the indicated cutoff of p.adj<0.1 is fine, but the final sentence of the Results subsection of the Abstract is unclear. It suggests that all three taxa achieved this significance threshold in the full cohort, but based on the Results and Table S1 only Parasutterella was significant across the full cohort (padj=10^-10), while all three taxa were significant specifically in UC (at a threshold of p.adj<0.05) but not CD in subset analyses. Please edit this sentence and possibly the Results section (under “Analysis of time (weeks) since treatment for single taxa“) to clarify the findings of the analysis on taxa associated with time elapsed since last treatment.  

We look forward to receiving your revised manuscript.

Kind regards,

Jonathan Jacobs

Academic Editor

PLOS ONE

Journal Requirements:

Additional Editor Comments:

The revised manuscript addressed all the reviewers’ initial comments. There was only a single comment by Reviewer #1 about the reporting of significant taxa associated with time elapsed since last infusion. I believe that the indicated cutoff of p.adj<0.1 is fine, but the final sentence of the Results subsection of the Abstract is unclear. It suggests that all three taxa achieved this significance threshold in the full cohort, but based on the Results and Table S1 only Parasutterella was significant across the full cohort (padj=10^-10), while all three taxa were significant specifically in UC (at a threshold of p.adj<0.05) but not CD in subset analyses. Please edit this sentence and possibly the Results section (under “Analysis of time (weeks) since treatment for single taxa“) to clarify the findings of the analysis on taxa associated with time elapsed since last treatment.

Reviewers' comments:

Reviewer's Responses to Questions

**Comments to the Author**

1. If the authors have adequately addressed your comments raised in a previous round of review and you feel that this manuscript is now acceptable for publication, you may indicate that here to bypass the “Comments to the Author” section, enter your conflict of interest statement in the “Confidential to Editor” section, and submit your "Accept" recommendation.

Reviewer #1: All comments have been addressed

Reviewer #2: All comments have been addressed

2. Is the manuscript technically sound, and do the data support the conclusions?

Reviewer #1: Yes

Reviewer #2: Yes

3. Has the statistical analysis been performed appropriately and rigorously? 

Reviewer #1: Yes

Reviewer #2: Yes

4. Have the authors made all data underlying the findings in their manuscript fully available?

Reviewer #1: Yes

Reviewer #2: Yes

5. Is the manuscript presented in an intelligible fashion and written in standard English?

Reviewer #1: Yes

Reviewer #2: Yes

6. Review Comments to the Author

Reviewer #1: Firstly, I would like to express my appreciation for the authors' thoughtful responses to my comments and suggestions. The revised paper is notably improved and more accessible compared to the original version. I have only one concern regarding the interpretation of the relationship between the taxa Parasutterella, Fusicatenibacter, and Anaerostipes with the duration since treatment in the CD group. Specifically, the distinction between statistical significance and nominal significance should be carefully maintained in both the abstract and discussion sections, as detailed below. The mention of an association with a p-value of less than 0.1, rather than the conventional threshold of less than 0.05, appears unconventional.

Secondly, I would like to sincerely apologize to the authors if the tone of some of my comments in the initial review seemed harsh; that was not my intention. I appreciate the thorough and constructive responses the authors provided despite this.

Although the article by Carlsen et al. has some limitations, these are thoroughly discussed within the text. Overall, the study adds valuable insight into the topic of the pIBD microbiome, particularly concerning the alpha diversity dynamics related to infliximab treatment.

Reviewer #2: (No Response)

7. PLOS authors have the option to publish the peer review history of their article (what does this mean?). If published, this will include your full peer review and any attached files.

Reviewer #1: No

Reviewer #2: No

---

## [Author Response · Author response to Decision Letter 1]

17 Sep 2024

Review Comments to the Author

 There was only a single comment by Reviewer #1 about the reporting of significant taxa associated with time elapsed since last infusion. I believe that the indicated cutoff of p.adj<0.1 is fine, but the final sentence of the Results subsection of the Abstract is unclear. It suggests that all three taxa achieved this significance threshold in the full cohort, but based on the Results and Table S1 only Parasutterella was significant across the full cohort (padj=10^-10), while all three taxa were significant specifically in UC (at a threshold of p.adj<0.05) but not CD in subset analyses. Please edit this sentence and possibly the Results section (under “Analysis of time (weeks) since treatment for single taxa“) to clarify the findings of the analysis on taxa associated with time elapsed since last treatment. 

Reply to reviewer:

We thank the reviewer very much for noticing this wording that can indeed be misleadning. We have changed the section and hope the reviewer agrees it now reflect the results accurately. The end of the subsection now reads:

At the genus level, a lower abundance of the genera Anaerostipes and Fusicatenibacter (Firmicutes), and a greater abundance of the genus Parasutterella (Proteobacteria), were associated (p.adj<0.05) with the time elapsed since last infusion in UC specifically, while only Parasutterella was associated across the full cohort (p.adj=1e-10).

As we have added a sentence to the section we needed to also shorten the abstract in other places and thus other adjustments have been made aiming to keep the meaning of the abstract intact. 

Comments with replies for reviewer Jakub Hurych, M.D., Ph.D.

Firstly we would like to thank the reviewer for this time and for the apology. While we indeed found the wording somewhat harsh, we do appreciate that written words can come across harsher than intented. No harm was done. 

Row 79 (and also 392-409 and 480-495)

1.“(p.adj<0.1) “ is no statistical association, this is only below 0.05.

There is no doubt about an association between the genera and the time since the last infusion in UC ( p.adj=0.037, 0.037, and 0.037; line 391). But for CD, there is no mention about statistical significance between the three genera and time since treatment, as line 395 shows a p-value <0.1 – it is only nominal and not statistical; I would mention it in the results, but discuss and highlight only the statistically significant results.

Similarilly, in the discussion, all is discussed as a clear significance between the three genera and time since treatment in both UC and CD. But again, this is not supported by the results. I would like to see Table 1 with all the modelling results, but I was not given the opportunity to see it (I asked via email, but with no response), unfortunately. But if the adjusted p-value for the relation between Parasuterella, Fusicatenibacter and Anaerostipes in CD is not below 0.05, it is not statistically significant and should be discussed as that.

To sum up, I would kindly ask the authors for either an explanation to or a correction in this regard.

Reply to reviewer:

This same point was made by reviewer 1 for the abstract and has been addressed as outlined above. For the result section of these taxa around lines 380-397 we only do call the associations significant that have p.adj<0.05 (being those in UC) and do spend some lines highlighting that the association differed between the two subtypes. We do mention other associations but do not call them significant. In the discussion section we have indeed missed to update the wording and thank the reviewer for noticing. It has not been updated to read: 

We also found a lower abundance of Anaerostipes (Firmicutes) and Fusicatenibacter (Firmicutes), and a higher abundance of the Proteobacteria Parasuterella, was significantly correlated with the time since the last IFX infusion in UC (p.adj<0.05) while only the association for Parasuterella was significant in the full cohort (p.adj=1e-10).

Row 97

It seems confusing: … “before diversity decreases again” - I got the point, but it took me a while; I would suggest the authors consider different wording to make the diversity dynamics statement clear enough.

Reply to reviewer:

We agree the wording was somewhat hard to follow. We have rephrased to now include the following: 

Our study illustrates that the composition of the microbiome is dynamic and alterable. Alpha diversity first increases rapidly (within the first four weeks) after treatment with infliximab, and then decreases again prior to the next infusion (at 8-12 weeks).

Row 425-6

I appreciate the idea of summarising the main point of the article, but it seems a word might be missing or I did not understood the meaning.

Reply to reviewer:

We agree that sentence was disconnected, and we have changed it to the following and hope it now hold the intended point for the discussion of the pattern observed for alpha diversity:

The observed diversity dynamics of this study likely reflect the concentration of IFX that will initially increase and then decrease over time after infusion, and consequently, the anti-inflammatory effect of the treatment, and thus the associated change in the microbiome, will depend on time since treatment.

---

## [Editor Report · Decision Letter 2]

23 Sep 2024

Gut microbiota diversity repeatedly diminishes over time following maintenance infliximab infusions in paediatric IBD patients

PONE-D-23-19986R2

Dear Dr. Thingholm,

We’re pleased to inform you that your manuscript has been judged scientifically suitable for publication and will be formally accepted for publication once it meets all outstanding technical requirements.

Kind regards,

Jonathan Jacobs

Academic Editor

PLOS ONE
---

## [Editor Report · Acceptance letter]

3 Oct 2024

PONE-D-23-19986R2 

PLOS ONE

Dear Dr. Thingholm, 

I'm pleased to inform you that your manuscript has been deemed suitable for publication in PLOS ONE. Congratulations! Your manuscript is now being handed over to our production team.

Kind regards, 

on behalf of

Dr. Jonathan Jacobs 

Academic Editor

PLOS ONE